# Discovery of Barakacin and Its Derivatives as Novel Antiviral and Fungicidal Agents

**DOI:** 10.3390/molecules28073032

**Published:** 2023-03-29

**Authors:** Yongyue Gao, Xingxing He, Lili Yan, Hongyu Zhang, Sijia Liu, Qian Ma, Peiyao Zhang, Yan Zhang, Zijun Zhang, Ziwen Wang, Aidang Lu, Qingmin Wang

**Affiliations:** 1Tianjin Key Laboratory of Structure and Performance for Functional Molecules, College of Chemistry, Tianjin Normal University, Tianjin 300387, China; 2School of Chemical Engineering and Technology, Hebei University of Technology, Tianjin 300401, China; 3State Key Laboratory of Elemento-Organic Chemistry, Research Institute of Elemento-Organic Chemistry, College of Chemistry, Collaborative Innovation Center of Chemical Science and Engineering (Tianjin), Nankai University, Tianjin 300071, China

**Keywords:** bisindole alkaloid, barakacin, anti-TMV activity, anti-fungal activity, mode of action

## Abstract

Pesticides are essential for the development of agriculture. It is urgent to develop green, safe and efficient pesticides. Bisindole alkaloids have unique and concise structures and broad biological activities, which make them an important leading skeleton in the creation of new pesticides. In this work, we synthesized bisindole alkaloid barakacin in a simple seven-step process, and simultaneously designed and synthesized a series of its derivatives. Biological activity research indicated that most of these compounds displayed good antiviral activities against tobacco mosaic virus (TMV). Among them, compound **14b** exerted a superior inhibitory effect in comparison to commercially available antiviral agent ribavirin, and could be expected to become a novel antiviral candidate. Molecular biology experiments and molecular docking research found that the potential target of compound **14b** was TMV coat protein (CP). These compounds also showed broad-spectrum anti-fungal activities against seven kinds of plant fungi.

## 1. Introduction

Since ancient times, agriculture has been the foundation of the national economy, and crop diseases and insect pests have been the main agricultural disasters perplexing the world. Their characteristics of wide distribution, irregular outbreak and serious harm have brought great difficulties to prevention and control. Pesticides are the key to plant protection and can effectively prevent and control the diseases of various crops [1]. However, the long-term and irregular use of traditional pesticides has caused serious environmental pollution, and also caused some pests to develop drug resistance. The development of novel, efficient and low-risk pesticides has become an important scientific issue that needs to be solved urgently. Natural products have novel structure, rich resources, extensive biological activities and a unique mechanism of action and are valuable resources for new pesticide discovery. Natural products can be made into pesticides directly or in the form of derivatives. They originate from nature and are used in nature, so they have ecological safety [2,3].

Indole ring system exists in a variety of natural products and has a wide range of pharmacological activities. Some of them have been used in various drugs, such as antihistamines [4], antifungals [5], antimicrobials [6], antioxidants [7], plant growth regulators [8], HIV inhibitors [9], anticonvulsants [10], anti-inflammatory analgesics [11], etc. Indole is probably one of the most important components of heterocyclic compounds for drug discovery [12]. Bisindole alkaloids are a class of star molecules in indole compounds. Since the isolation of a bis(indolyl)imidazole topsentin and its analogs from *Topsentia genitrix* more than a decade ago [13], scientists have successively isolated various novel bisindole alkaloids from marine organisms as well as from microorganisms; they found that these alkaloids had powerful and diverse biological activities, including antiviral, antitumor, antibacterial and anti-inflammatory activities [14]. Their unique and concise structures and broad biological activities make them an important leading skeleton in the creation of new drugs.

Bisindole alkaloid barakacin was first isolated from the ruminal bacterium *Pseudomonas aeruginosa* strain ZIO in 2012 and found to have weak and non-selective cytotoxic activity against human cancer cell lines LXFA 629L, LXFL 529L (lung), MAXF 401NL (breast), etc. [15]. In 2016, Buzid’ s group accomplished the first synthesis of barakacin with an overall yield of 11% (Appendix A) [16]. So far, the research on barakacin is limited to this, and there is no report on the control of crop diseases and insect pests.

Tobacco mosaic virus (TMV) has a history of more than 100 years, and with the development of the first half of the 20th century, TMV has become the preferred model virus for basic research such as disease resistance and breeding [17,18]. After the plant is infected, the virus proliferates rapidly on the leaves, causing extensive damage to the leaves and a large number of necrotic spots on the surface. The severely diseased leaves will shrink, twist or even die. At present, there is no specific drug for TMV. Ribavirin, the widely used antiviral agent, can only give an inhibitory effect of less than 50% at the concentration of 500 µg/mL.

We have been engaged in the discovery of new antiviral agents based on natural products and found a series of alkaloids that had good antiviral activities [19,20,21]. In this work, we chose the alkaloid barakacin as the lead compound, designed and synthesized a series of its derivatives, evaluated their antiviral activities against TMV and investigated their structure-activity relationship and antiviral action mechanism systematically (Figure 1). In addition, the fungicidal activities of these compounds against seven phytopathogenic fungi (*Fusarium oxysporium f. sp. cucumeris*; *Physalospora piricola*; *Rhizoctonia cerealis*; *Alternaria solani*; *Botrytis cinereal*; *Phytophthora capsici*; *Sclerotinia sclerotiorum*) were also investigated.

## 2. Results and Discussion

### 2.1. Chemistry

Buzid’s group achieved the synthesis of alkaloid barakacin with an overall yield of 11%, but the yield of the Suzuki coupling step was only 26%, and the raw materials were not easy to obtain, which limited its application in the study of structure-activity relationship (SAR) of these compounds [16]. Therefore, it is necessary to develop a simple and practical route to realize the structural diversity derivation of these compounds. Using the commercially available *o*-hydroxybenzylnitrile as raw material, compound **1** was obtained by methylation, which reacted with sodium hydrosulfide to produce thioamide **2**. Compound **2** reacted with ethyl bromopyruvate to obtain thiazole **3**, and then the key intermediate aldehyde **5** was obtained through reduction and oxidation of **3**. Aldehyde **5** reacted with indole under the catalysis of I_2_ to give alkaloid derivative **6**. After compound **6** was demethylated, the natural product barakacin (**7**) was obtained via seven steps in 26% overall yield (Figure 1). Compared with the methods reported in the literature, the raw materials of our route are easy to obtain, the operation is simple and it is easier to realize the structural diversity derivation of these kinds of compounds. In order to evaluate the effects of hydroxyl on benzene ring and indole NH on biological activity, we designed and synthesized derivatives **8a**, **8b** and **9a**–**9e** (Figure 1). To explore the influence of different substituents on the indole benzene ring on the biological activity, compounds **13a**–**13d** were designed and synthesized in a similar way (Figure 2). Heterocycles are important pharmacophores with a wide range of biological activities. Based on structural simplification and scaffold hopping strategy, we designed and synthesized bisindole derivatives **14a**–**14g** with different ring structures (Figure 3). To further investigate the substitution effect of the heterocyclic region, we designed and synthesized derivatives **15** and **16a**–**16c** (Figure 4).

### 2.2. Antiviral Activity Result and Structure-Activity Relationship (SAR)

Alkaloid barakacin (**7**) and its derivatives **6**, **8a**, **8b**, **9a**–**9e**, **13a**–**13d**, **14a**–**14g**, **15**, **16a**–**16c** were screened for their anti-TMV activities with commercial ribavirin as control. Results in Table 1 show that most of these compounds displayed very good in vivo antiviral activities. Among them, compounds **8a**, **9d**, **13b**, **13d**, **14a**, **14c**, **14d** and **14g** presented similar levels of antiviral activities as ribavirin. The antiviral activities of natural compound barakacin (**7**) and its derivatives **8b**, **9c**, **13a**, **13c** and **14b** are better than that of ribavirin. Compound **14b**, with advantages of simple structure, convenient synthesis and good biological activity, emerged as a new antiviral candidate.

Barakacin (**7**) displayed prominent antiviral activity; the introduction of methyl or propargyl into the hydroxyl of phenyl ring leads to the decrease in activity (inhibition rate: **7** > **8a** > **6**), while the introduction of benzyl can maintain activity (inhibition rate: **7** ≈ **8b**). Indole NH also has a great impact on the antiviral activity: the introduction of benzyl or ethylsulfonyl into indole NH leads to the decrease in activity (inhibition rate: **9b** ≈ **9e** < **6**); the introduction of methyl can maintain the original activity (inhibition rate: **9a** ≈ **6**); and the introduction of *p*-toluenesulfonyl or benzoyl can improve the activity (inhibition rate: **9c** > **9d** > **6**), indicating that the introduction of electron-withdrawing aryl in this region is beneficial to the antiviral activity. Compounds **13a**–**13d** have the same level of inhibitory activity, indicating that the introduction of functional groups at position 5 or 6 of indole ring is tolerable. The structure simplification of natural products has always been an important strategy for the creation of new pesticides [22,23]. It can not only save the synthesis costs, but also improve the physical and chemical properties of natural products. Based on structural simplification and scaffold hopping strategy, compounds **14a**–**14g** were synthesized by one-step reaction from commercially available indole and corresponding aldehydes. Except for **14e** and **14f**, all compounds showed similar or higher level of antiviral activities as ribavirin, especially compound **14b**, which exhibited similar biological activity to parent compound **7**. Taking **14c** as an example, we further investigated the substitution effect on NH. The results showed that the introduction of methyl, benzyl or *p*-toluenesulfonyl on NH was not conducive to biological activity (inhibition rate: **14c** > **15** and **16a**–**16c**).

### 2.3. Preliminary Mechanism Research

TMV is a baculovirus with a total length of 300 nm, which is composed of CP and RNA; usually, protein monomers first form 20S disk, and then assemble with RNA into complete virus particles; in this process, CP plays a very important role [24]. We selected compound **14b** to further evaluate the mode of action of these compounds via our reported method [20] via transmission electron microscopy (FEI Tecnai G2 F20). Results in Figure 2 show that TMV CP and RNA could assemble into virus particles under conventional conditions, but the assembly was blocked when compound **14b** was added, indicating that compound **14b** can interfere with the assembly process of the virus. Considering the good inactive activity of compound **14b**, we speculated that it might act on the CP. We further designed the interaction experiment between **14b** and TMV protein (Figure 3). The results show that under normal circumstances, the protein monomer could successfully generate the 20S disk with a regular shape, but when **14b** was added, the formation of the 20S disk could hardly be observed, indicating that the compound could act on the viral protein. From the preliminary mechanism research, it can be inferred that these compounds inhibit virus assembly by acting on viral proteins.

### 2.4. Molecular Docking

Molecular docking is an important and advantageous tool to verify the interaction sites and affinities between biological macromolecules and small molecule drugs at the molecular level. It is widely used in new drug creation and target prediction [25,26,27]. In order to evaluate the hypothetical binding mode between compounds **8b**, **9a** and **14b** with TMV CP (PDB code: 1EI7), we carried out molecular docking via AutoDock Vina 1.1.2 [28]. As shown in Figure 4: compound **8b** and amino acid residue SER-138 formed a 2.1 Å hydrogen bond (Figure 4A); compound **14b** formed a 3.2 Å hydrogen bond with amino acid residues GLU-131 and PRO-254, respectively (Figure 4C); while compound **9a** with poor activity did not form a hydrogen bond with the target protein (Figure 4B). The results of molecular docking are consistent with SAR, which further confirms the interaction between these compounds and TMV CP.

### 2.5. Fungicidal Activity Result and Structure-Activity Relationship (SAR)

Considering the biodiversity of natural products, we further studied the antifungal activities of these compounds with commercial fungicides carbendazim and chlorothalonil as controls (Table 2). The results showed that the target compounds also exhibited broad-spectrum fungicidal activities, among which compounds **6**, **7**, **8a**, **13a**, **14a**–**14c**, **14e** and **14g** displayed better inhibitory activities against *Botrytis cinereal* than chlorothalonil. The anti-fungal activities of **8a**, **9a**–**9e**, **14b**, **16b** and **16c** against *Sclerotinia sclerotiorum* are superior to those of commercial fungicides carbendazim and chlorothalonil. The fungicidal activity of **14b** against *Rhizoctonia cerealis* is better than that of carbendazim. In general, these compounds showed good inhibitory activities against *Physalospora piricola* and *Sclerotinia sclerotiorum*. For *Sclerotinia sclerotiorum*: barakacin (**7**) presented a similar level of fungicidal activity as carbendazim; the introduction of propargyl group into the hydroxyl group of phenyl ring can improve the fungicidal activity; compounds **9a**–**9e** exhibited better activities than **6**, indicating that the introduction of substituents into indole NH can improve the fungicidal activity; compound **13a** showed higher fungicidal activity than **13b**–**13d**, indicating that the introduction of functional groups at position 5 or 6 of the indole ring is unfavorable to the activity; except for compound **14f**, all compounds with simplified structure (**14a**–**14e**, **14g**, **15**, **16a**–**16c**) showed fungicidal activities equivalent to or better than that of natural product **7**. Compound **14b** also displayed broad-spectrum and efficient fungicidal activity and could be used as novel anti-fungal candidate.

## 3. Materials and Methods

### 3.1. Synthetic Procedures

#### 3.1.1. Chemicals

The reagents were purchased from commercial sources and were used as received. All anhydrous solvents were dried and purified by standard techniques prior to use. The synthesis steps of derivatives **8a**–**8b**, **9a**–**9e**, **10**–**12**, **13a**–**13d**, **14a**–**14g**, **15**, **16a**–**16c** are placed in Appendix A.

#### 3.1.2. Instruments

Melting points were determined on an X-4 melting point apparatus equipped with a binocular microscope (Beijing Tech Instruments Co., Beijing. China). Nuclear magnetic resonance (NMR) spectra were obtained with a 400 MHz (100 MHz for ^13^C) instrument (Bruker, Billerica, MA, USA) at room temperature in either CDCl_3_ or DMSO-*d*_6_ as the solvent. Chemical shifts were measured relative to residual solvent peaks of DMSO-*d*_6_ as internal standards (^1^H: *δ* = 2.5 and 3.3 ppm; ^13^C: *δ* = 39.9 ppm) or CDCl_3_ as internal standards (^1^H: *δ* = 7.26 ppm; ^13^C: *δ* = 77.0 ppm). The following abbreviations are used to designate chemical shift multiplicities: s = singlet, d = doublet, dd = doublet of doublets, t = triplet, m = multiplet and brs = broad singlet. High-resolution mass spectra were obtained with an Ionspec, 7.0 T Fourier transform ion cyclotron resonance mass spectrometer.

#### 3.1.3. Preparation of 2-Methoxybenzonitrile (**1**)

The 2-Cyanophenol (1.00 g, 8.4 mmol) and K_2_CO_3_ (1.16 g, 8.4 mmol) were dissolved in 50 mL of acetone, and CH_3_I (1.05 mL, 16.8 mmol) was added dropwise to the mixture at 0 °C. The progress of the reaction was monitored by thin layer chromatography (TLC). After the reaction was complete, the solvent was removed in vacuo at 50 °C, water was added, the reaction solution was extracted with ethyl acetate (50 mL × 3), dried with anhydrous Na_2_SO_4_, filtered with suction and the solvent was removed in vacuo at 50 °C to obtain **1** as a yellow oil with 95% yield; ^1^H NMR (400 MHz, CDCl_3_) δ 7.54 (ddd, *J* = 7.4, 4.4, 2.6 Hz, 2H), 7.00 (ddd, *J* = 12.2, 6.5, 2.7 Hz, 2H), 3.93 (s, 3H). ^13^C NMR (100 MHz, CDCl_3_) δ 161.2, 134.5, 133.7, 120.8, 116.6, 111.4, 101.6, 56.0.

#### 3.1.4. Preparation of 2-Methoxybenzothioamide (**2**)

An amount of 70% NaHS (3.00 g, 52.57 mmol) and MgCl_2_·6H_2_O (5.34 g, 26.29 mmol) were added to 20 mL of DMF, and compound **1** dissolved in 10 mL of DMF was added dropwise to the system under vigorous stirring. The reaction was stirred at 40 °C for 6 h. After the reaction was completed by TLC monitoring, part of the solvent was removed, 100 mL of 1 M hydrochloric acid was added, and the mixture was stirred for 0.5 h. The reaction solution was extracted with ethyl acetate (30 mL × 3), the organic layer was washed with water, dried over anhydrous Na_2_SO_4_, and concentrated to obtain compound **2** as a reddish-brown solid with 87% yield, mp. 139–141 °C. ^1^H NMR (400 MHz, CDCl_3_) δ 9.05 (s, 1H), 8.63 (dd, *J* = 8.0, 1.8 Hz, 1H), 8.18 (s, 1H), 7.47 (ddd, *J* = 8.4, 7.3, 1.8 Hz, 1H), 7.20–6.87 (m, 2H), 3.97 (s, 3H). ^13^C NMR (100 MHz, CDCl_3_) δ 199.2, 156.3, 136.4, 133.7, 124.9, 121.3, 111.5, 56.1.

#### 3.1.5. Preparation of Ethyl 2-(2-Methoxyphenyl)thiazole-4-carboxylate (**3**)

Compound **2** (2.00 g, 12 mmol) was dissolved in 200 mL of ethanol, and under stirring at 0 °C, ethyl 3-bromo-2-oxopropionate (3.50 g, 18 mmol) was gradually added dropwise to the system. Then, the ice bath was removed, and the reaction system was heated to 90 °C. After the completion of the reaction by TLC monitoring, the solvent was evaporated under reduced pressure to obtain a crude product, which was purified by flash chromatography on silica gel using petroleum ether and ethyl acetate (4:1, *v*/*v*) as eluent to give title compound **3** as a pale yellow solid with 92% yield, mp. 137–139 °C; ^1^H NMR (400 MHz, CDCl_3_) δ 8.45 (dd, *J* = 7.9, 1.7 Hz, 1H), 8.13 (s, 1H), 7.50–7.27 (m, 1H), 7.09–6.86 (m, 2H), 4.37 (q, *J* = 7.1 Hz, 2H), 3.96 (s, 3H), 1.36 (t, *J* = 7.1 Hz, 3H). ^13^C NMR (100 MHz, CDCl_3_) δ 163.0, 162.0, 156.5, 146.0, 131.4, 129.1, 127.9, 121.5, 121.1, 111.2, 61.3, 55.6, 14.4.

#### 3.1.6. Preparation of (2-(2-Methoxyphenyl)thiazol-4-yl)methanol (**4**)

LiAlH_4_ (0.14 g, 3.8 mmol) was dissolved in 20 mL of THF, and under stirring at 0 °C, it was slowly dropped into a solution of compound **3** (1.00 g, 3.8 mmol) in THF. The mixture was stirred at room temperature and monitored by TLC. After the reaction was completed, water was added to quench it, the result solution was extracted with ethyl acetate, dried over anhydrous Na_2_SO_4_, and concentrated to obtain compound **4** as a yellow solid with 85% yield, mp. 116–118 °C; ^1^H NMR (400 MHz, CDCl_3_) δ 8.40 (dd, *J* = 7.8, 1.7 Hz, 1H), 7.54–7.35 (m, 1H), 7.39 (s, 1H), 7.19–6.99 (m, 2H), 4.86 (s, 2H), 4.04 (s, 3H), 2.95 (s, 1H). ^13^C NMR (100 MHz, CDCl_3_) δ 162.9, 156.4, 154.9, 130.9, 128.4, 121.9, 121.1, 115.5, 111.4, 61.1, 55.6.

#### 3.1.7. Preparation of 2-(2-Methoxyphenyl)thiazole-4-carbaldehyde (**5**)

Under stirring at room temperature, PCC (1.46 g, 6.8 mmol) was dissolved in 45 mL of dichloromethane, and a solution of compound **4** (1.00 g, 4.5 mmol) in dichloromethane (40 mL) was added dropwise to it. After 4 h of reaction, the mixture was washed with 10% hydrochloric acid (30 mL × 2) and extracted with ethyl acetate (50 mL × 3). The organic layer was washed with brine (50 mL), water (50 mL × 2), then dried over anhydrous Na_2_SO_4_, filtered by suction, and concentrated in vacuo. The residue was purified by flash chromatography on silica gel using petroleum ether and ethyl acetate (4:1, *v*/*v*) as eluent to obtain the target compound **5** as a white solid with 81% yield, mp. 81–83 °C; ^1^H NMR (400 MHz, CDCl_3_) δ 10.14 (s, 1H), 8.49 (dd, *J* = 7.9, 1.7 Hz, 1H), 8.22 (s, 1H), 7.63–7.44 (m, 1H), 7.11 (ddd, *J* = 22.3, 14.6, 4.7 Hz, 2H), 4.06 (s, 3H). ^13^C NMR (100 MHz, CDCl_3_) δ 185.4, 163.7, 156.6, 153.7, 131.7, 128.8, 128.0, 121.3, 111.4, 55.6.

#### 3.1.8. Preparation of 4-(Di(1*H*-indol-3-yl)methyl)-2-(2-methoxyphenyl)thiazole (**6**)

Indole (0.468 g, 4 mmol) and compound **5** (0.219 g, 1 mmol) were dissolved in acetonitrile (45 mL), and I_2_ (0.127 g, 0.5 mmol) was added with stirring. After reacting for 30 min at room temperature, the reaction was quenched by adding 5% Na_2_S_2_O_3_ solution (20 mL). The organic solvent was removed in vacuo at 55 °C, and the resulting solution was extracted with ethyl acetate (50 mL × 3). Then, the organic phase was washed with brine (100 mL), dried over anhydrous Na_2_SO_4_, filtered with suction and concentrated. The residue was purified by column chromatography with petroleum ether and ethyl acetate (5:1, *v*/*v*) as eluent to obtain compound **6** as a pale yellow solid with 71% yield, mp. 210–212 °C; ^1^H NMR (400 MHz, DMSO-*d*_6_) δ 10.82 (d, *J* = 1.8 Hz, 2H), 8.24 (dd, *J* = 7.9, 1.7 Hz, 1H), 7.45 (d, *J* = 8.2 Hz, 2H), 7.44–7.40 (m, 1H), 7.33 (d, *J* = 8.1 Hz, 2H), 7.29 (s, 1H), 7.22 (d, *J* = 8.0 Hz, 1H), 7.09–7.00 (m, 5H), 6.88 (dd, *J* = 11.0, 3.9 Hz, 2H), 6.07 (s, 1H), 3.99 (s, 3H). ^13^C NMR (100 MHz, DMSO-*d*_6_) δ 160.7, 159.2, 156.4, 136.9, 131.2, 128.1, 127.0, 124.0, 122.1, 121.3, 119.6, 118.5, 117.5, 116.5, 112.7, 111.9, 56.3, 36.9. HRMS (ESI) calcd for C_27_H_22_N_3_OS [M+H]^+^ 436.1478, found 436.1472.

#### 3.1.9. Preparation of Barakacin (**7**)

Compound **6** (1.00 g, 2.3 mmol) was dissolved in dichloromethane, and a solution of 1 mol/L BBr_3_ in CH_2_Cl_2_ (28 mL, 27.6 mmol) was added dropwise at −78 °C under an inert atmosphere. The reaction solution was stirred at room temperature overnight, and then H_2_O (28 mL, 27.6 mmol) was added. The reaction solution was partitioned and extracted with ethyl acetate (30 mL × 3), the organic phase was washed with brine (100 mL), dried over anhydrous Na_2_SO_4_, filtered with suction and concentrated. The residue was subjected to column chromatography with petroleum ether and ethyl acetate (2:1, *v*/*v*) as eluent to obtain compound **7** as a red solid with 70% yield, mp. 118–120 °C; ^1^H NMR (400 MHz, DMSO-*d*_6_) δ 11.71 (s, 1H), 10.87 (s, 2H), 8.00–7.79 (m, 1H), 7.42 (dd, *J* = 24.7, 5.9 Hz, 2H), 7.43–7.22 (m, 4H), 7.15–6.98 (m, 4H), 6.99–6.84 (m, 4H), 6.09 (d, *J* = 11.8 Hz, 1H). ^13^C NMR (100 MHz, DMSO-*d*_6_) δ 165.0, 158.3, 155.4, 136.4, 131.1, 127.2, 126.5, 123.4, 120.9, 119.5, 119.0, 118.3, 116.8, 116.5, 114.6, 111.5, 36.1. HRMS (ESI) calcd for C_26_H_18_N_3_OS [M-H]^−^ 420.1176, found 420.1179.

### 3.2. Biological Assay

Each test was repeated three times at 25 ± 1 °C. The active effect is expressed in percentage scale of 0–100 (0: no activity; 100: total inhibited).

Specific steps for the anti-TMV activity and fungicidal activity tests and mode of action research were carried out using the literature method [19,20], which also can be seen in Appendix A.

## 4. Conclusions

In summary, using cheap o-hydroxybenzonitrile as raw material, the natural product barakacin was synthesized in seven steps with a total yield of 26%, which provided a practical method for obtaining the structural diversity derivatives of these compounds. A series of barakacin derivatives were designed based on structural simplification and scaffold hopping strategy, synthesized, and evaluated for their antiviral activities. These compounds were found to have good anti-TMV activities for the first time. Through systematic SAR research, compound **14b** was found to be a new antiviral candidate with advantages of simple structure, convenient synthesis and good biological activity. Mode of action research revealed that these compounds might inhibit virus assembly by acting on viral proteins. Further fungicidal activity research indicated that these compounds also exerted broad-spectrum anti-fungal activities, especially for *Physalospora piricola* and *Sclerotinia sclerotiorum*. Compound **14b** with broad-spectrum and efficient fungicidal activity emerged as a novel anti-fungal candidate. This work provides a case for the discovery of new pesticides based on the structural simplification of natural products.

## Data Availability

All data used to support the findings of this study are included within the article and Appendix A.

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
