# Peer review of "Discovery of Barakacin and Its Derivatives as Novel Antiviral and Fungicidal Agents"

_molecules, 2023, doi:10.3390/molecules28073032_

Round 1

Reviewer 1 Report

The manuscript by Gao et al titled, " Discovery of Barakacin and Its Derivatives as Novel Antiviral and Fungicidal Agents" provides valuable information about novel anti-viral and anti-fungal compounds. The authors employ various techniques to verify the efficacy of the purified compounds. The methods seem to be executed with scientific rigor and the results are communicated in a comprehensible manner. However, I have a few major and minor comments for the authors to pay heed to.

Major comments:

1. I think the authors should add some images of the antifungal assay in the Results or in the supplementary information. Though not necessary still the images might be of interest to future readers.

2. In the Methods section, the methodology describing each compound is extensive and a bit monotonous. I think the authors can maybe describe a few compounds (of interest or that had good antiviral/or antifungal activity) and provide details of the other compounds in the Supplementary information. I leave it to the authors how they wish to deal with this issue.

Minor comments

• Line 44, replace “modified” with “used”.                                                                    • Line 50, start “Scientists” as a new sentence. In its current form that is a very long sentence to understand for readers.                                                          • Line 71, add “that” before “had”.                                                                                  • Line 115, kindly rephrase “good to excellent”.                                                        • I think the full name of the fungal plant pathogens should be given in the text.

Author Response

Dear Editors and Reviewers.

   Thank you for your letter and comments concerning our manuscript entitled “Discovery of Barakacin and Its Derivatives as Novel Antiviral and Fungicidal Agents” (Manuscript ID: molecules-2264099). We have studied comments carefully and have made correction which we hope meet with approval.  

  As the Journal requirements: The revised manuscript were uploaded two versions: one version as a clean copy, another version marked yellow background for all changes.

 Reviewer #1:

  1. I think the authors should add some images of the antifungal assay in the Results or in the supplementary information. Though not necessary still the images might be of interest to future readers.

Response: Thanks very much for your good comments. Adding pictures to the article can indeed increase readability, but during the testing of this batch of samples, the testers did not provide corresponding pictures. We will prepare in advance for the following work. Thank you. 

  1. In the Methods section, the methodology describing each compound is extensive and a bit monotonous. I think the authors can maybe describe a few compounds (of interest or that had good antiviral/or antifungal activity) and provide details of the other compounds in the Supplementary information. I leave it to the authors how they wish to deal with this issue.

Response: Thanks very much for your good comments. The total synthesis process of natural product Barakacin (7) remains in the text, and the synthesis steps of derivatives 8a8b, 9a9e, 1012, 13a13d, 14a14g, 15, 16a16c are transferred to SI.

  1. Minor comments
  • Line 44, replace “modified” with “used”.                                                                    • Line 50, start “Scientists” as a new sentence. In its current form that is a very long sentence to understand for readers.                                                          • Line 71, add “that” before “had”.                                                                                  • Line 115, kindly rephrase “good to excellent”.                                                        • I think the full name of the fungal plant pathogens should be given in the text.

Response: Thanks very much for your good comments. The above errors have been corrected.

We appreciate for Editors and Reviewers’ warm work earnestly, and hope that the corrections will meet with approval.

Once again, thank you very much for your comments and suggestions.

We look forward to your information about my revised papers and thank you for your good comments.    

Yours sincerely,

Ziwen Wang

Reviewer 2 Report

The authors presented a study devoted to the development of new antiviral and antifungal compounds for plant protection based on Barakacin. The development of new plant protection products is an important direction, since the protection of crops from diseases allows both to preserve the yield and increase it. The authors presented a seven-step synthesis of Barakacin with a total yield of 26%. The authors also synthesized a number of derivatives of barakacin. This paper can be recommended for publication in Molecules. However, there are a few comments:

Line 85: Please add the scheme for the synthesis of Barakacin by Buzid’s group.

Line 121: “displayed excellent antiviral activity”. Researchers are always full of optimism and try to achieve the maximum result. In this case, "excellent" is premature. The data suggests moderate activity.

Lines 179-189: This paragraph is a duplicate of the previous one.  

Authors should add HRMS spectra to supplementary materials

13C NMR spectrum of 2: please adjust the baseline.

Author Response

Dear Editors and Reviewers.

   Thank you for your letter and comments concerning our manuscript entitled “Discovery of Barakacin and Its Derivatives as Novel Antiviral and Fungicidal Agents” (Manuscript ID: molecules-2264099). We have studied comments carefully and have made correction which we hope meet with approval.  

  As the Journal requirements: The revised manuscript were uploaded two versions: one version as a clean copy, another version marked yellow background for all changes.

Reviewer #2:

  1. Line 85: Please add the scheme for the synthesis of Barakacin by Buzid’s group.

Response: Thanks very much for your good comments. The scheme for the synthesis of Barakacin by Buzid’s group has been added into SI (Scheme S1).

  1. Line 121: “displayed excellent antiviral activity”. Researchers are always full of optimism and try to achieve the maximum result. In this case, "excellent" is premature. The data suggests moderate activity.

Response: Thanks very much for your good comments. Excellent has been changed. In addition, plant viral diseases are known as plant cancers and are difficult to control. Currently, the most effective antiviral agent, ningnanmycin, can only provide 56% of the control effect. Therefore, the antiviral activities of these compounds are still prominent.

  1. Lines 179-189: This paragraph is a duplicate of the previous one.  

Response: Yes, thanks, the error has been corrected.

  1. Authors should add HRMS spectra to supplementary materials

Response: Thank you very much for your suggestions. The spectra of HRMS provided by our test platform were printed paper versions. The author who collected spectra felt that many journals did not need to provide copies of HRMS spectra, so he did not scan the printed paper versions into an electronic version for storage in time. It is our negligence. At present, he has graduated and left school, and have failed to find the printed paper versions of HRMS spectra. I hope you could understand.

  1. 13C NMR spectrum of 2: please adjust the baseline.

Response: Thank you very much for your suggestions. 13C NMR spectrum of 2 has been adjusted.

We appreciate for Editors and Reviewers’ warm work earnestly, and hope that the corrections will meet with approval.

Once again, thank you very much for your comments and suggestions.

We look forward to your information about my revised papers and thank you for your good comments.    

Yours sincerely,

Ziwen Wang